# Effect of Plasma Exchange Treatment in Patients with Hypertriglyceridemia-Induced Acute Pancreatitis

**DOI:** 10.3390/medicina59050864

**Published:** 2023-04-29

**Authors:** Duy Cuong Nguyen, Ngoc Anh Nguyen, Quang Kien Dinh, Dinh Tuan Le, Thanh Binh Vu, Van Thuan Hoang

**Affiliations:** 1Thai Binh University of Medicine and Pharmacy, Thai Binh 410000, Vietnam; 2Military Hospital 103, Vietnam Military Medical University, Hanoi 12110, Vietnam

**Keywords:** hypertriglyceridemia, triglyceride, pancreatitis, plasma exchange

## Abstract

*Background and Objectives*: To describe the clinical and biological characteristics of patients with hypertriglyceridemia-induced acute pancreatitis (HTG-AP) and to evaluate the effectiveness of therapeutic plasma exchange (TPE) in the management of HTG-AP. *Materials and Methods*: A cross-sectional study was conducted on 81 HTG-AP patients (30 treated with TPE and 51 treated conventionally). The main outcome was a decrease in serum triglyceride levels (<11.3 mmol/L) within 48 h of hospitalization. *Results*: The mean age of participants was 45.3 ± 8.7 years, and 82.7% were male. Abdominal pain was the most frequent clinical sign (100%), followed by dyspepsia (87.7%), nausea or vomiting (72.8%), and a bloated stomach (61.7%). The HTG-AP patients treated with TPE had significantly lower calcemia and creatinemia levels but higher triglyceride levels than those who received conservative treatment. They also had more severe diseases than those treated conservatively. All patients in the TPE group were admitted to the ICU, whereas the ICU admission rate in the non-TPE group was 5.9%. The TPE patients were more likely to experience a rapid decrease in triglyceride levels within 48 h of treatment than those treated conventionally (73.3% vs. 49.0%, *p* = 0.03, respectively). The decrease in triglyceride levels did not depend on the age, gender, or comorbidities of the HTG-AP patients or the severity of disease. However, TPE and early treatment in the first 12 h of disease onset were effective in rapidly reducing serum triglyceride levels (adjusted OR = 3.00, *p* = 0.04 and aOR = 7.98, *p* = 0.02, respectively). *Conclusions*: This report demonstrates the effectiveness of early TPE in reducing triglyceride levels among HTG-AP patients. More randomized clinical trials studies with a large sample size and post-discharge follow-up are needed to confirm the effectiveness of TPE methods in managing HTG-AP.

## 1. Introduction

Acute pancreatitis (AP) is the acute, diffuse inflammation of the pancreatic parenchyma and can vary in severity. People with mild and moderate AP usually recover and are discharged within a short time and generally have a good prognosis [1]. However, severe AP can cause complications such as multi-organ failure, abdominal bleeding, and peritonitis. Previous studies have shown that about 20% of severe cases progress to necrotizing AP, with a mortality of 30–40% despite aggressive treatment [1,2,3]. In addition, patients with AP can experience many recurrences after being discharged from hospital, affecting the patient’s health and quality of life. AP is also a risk factor for pancreatic cancer and diabetes [4,5].

Gallstones and alcoholism are known to be the leading causes of AP. However, hypertriglyceridemia is also an important and increasingly common cause in recent years [2]. In particular, rates of AP with hypertriglyceridemia are increasing due to dyslipidemia, which is related to the growing prevalence of overweight and obesity in the community [6]. Nevertheless, the pathophysiological mechanism of AP due to hyperlipidemia is not yet fully understood. AP with hypertriglyceridemia is dependent not only on the inflammatory response induced by the pancreas but also on the lipid toxic injury from triglyceride hydrolysis. This can be explained by two main mechanisms: the formation and increased concentration of chylomicrons and the breakdown of triglycerides into free fatty acids in the pancreas [7,8,9]. Indeed, chylomicrons are lipoprotein particles that transport dietary fat from the intestines to other parts of the body. When their production rate exceeds the rate of clearance, they accumulate in the blood and can cause tissue damage, including in the pancreas. The breakdown of triglycerides into free fatty acids in the pancreas can also lead to tissue injury by promoting the formation of reactive oxygen species and triggering an inflammatory response [7,9].

Patients with high blood triglyceride levels are at a high risk of developing AP. Furthermore, compared to other causes of AP, hypertriglyceridemia-induced acute pancreatitis (HTG-AP), particularly with triglyceride levels above 5.6 mmol/L, is associated with greater clinical severity and a high rate of complications [10]. Hypertriglyceridemia can also lead to pancreatic microvascular disease, impairing the flow of blood to the pancreas and contributing to tissue injury [10]. This can result in a more severe form of AP and an increased risk of complications, such as pancreatic necrosis and infected pancreatic necrosis [10].

HTG-AP has historically been associated with a poor prognosis. It is currently receiving more attention with advanced treatments and the prognosis is usually good. While there are currently no clear evidence-based guidelines for the definitive management of HTG-AP [11], reducing triglyceride levels as early as possible has become a common therapeutic strategy. Therapeutic plasma exchange (TPE) is being applied to quickly bring the patient’s blood triglyceride level below 11.3 mmol/L, especially after the first session [8,12,13,14]. Previous studies have demonstrated the effectiveness of TPE in the management of HTG-AP [15]. In a retrospective study of patients with severe HTG-AP, TPE was found to significantly reduce serum triglyceride levels and improve clinical outcomes [16]. Chen et al. conducted a retrospective case–control study among 46 patients receiving TPE and 135 patients undergoing conservative treatments and showed that TPE resulted in a short-term and rapid reduction in plasma triglyceride concentrations [12]. In addition to its effectiveness in reducing triglyceride levels, TPE has also been shown to have a favorable safety profile in patients with HTG-AP [14]. A literature review found that the best clinical benefit concerning reductions in morbidity and mortality can be achieved when TPE is used as early as possible [17]. Additionally, in patients with severe HTG-AP who underwent TPE, no serious adverse events related to the procedure were reported [17].

It should be noted that the optimal timing and frequency of TPE in the management of HTG-AP is still a subject of debate. Some studies have suggested that TPE should be initiated as early as possible, with multiple sessions performed over the course of several days [14,18]. Another study suggested that a single TPE session may be sufficient to achieve significant reductions in triglyceride levels [19]. Further research is needed to determine the most effective TPE protocol for the management of HTG-AP.

In Vietnam, recent studies have shown that the incidence of AP is increasing with socioeconomic development, particularly HTG-AP [14,20]. The rise in HTG-AP cases is attributed to the changes in lifestyle and dietary habits that have accompanied the country’s rapid economic growth, as well as the increased prevalence of obesity, which is a known risk factor for HTG-AP [21]. The consumption of alcohol and smoking, which are also risk factors for AP, have also increased among the Vietnamese population [22], further contributing to the increase in AP cases. HTG-AP is associated with significant morbidity and mortality and its management can be challenging, underscoring the need for effective and evidence-based treatment options. In the treatment of HTG-AP, the plasma exchange method is being applied in many central and regional hospitals, helping to rapidly improve the prognosis of the disease and save the lives of several patients [14]. However, the studies conducted to objectively evaluate the effectiveness of this treatment in Vietnam are still very limited. Therefore, we conducted this study with the aim of describing the clinical and biological characteristics of patients with HTG-AP and evaluating the effectiveness of TPE in the management of HTG-AP.

## 2. Materials and Methods

### 2.1. Study Design and Participants

This was a descriptive cross-sectional study conducted at the Intensive Care Unit (ICU) and Gastroenterology Department of Thai Binh General Hospital from January 2020 to December 2021. All adult patients admitted with HTG-AP during the period of time of the study were selected. Once the diagnosis was confirmed, the patients and their family members were informed about TPE and possible complications. They had the right to accept or refuse the treatment. If they agreed to participate, they were placed in the TPE group (in addition to plasma exchange, these patients also received conventional treatment); otherwise, they were placed in the non-TPE group (conventional treatment only). Patients with AP due to other factors, such as alcohol, trauma, cholelithiasis, or neoplasm, those who refused to participate in the study or were transferred to another hospital before treatment, and pregnant women with HTG-AP were excluded from our study.

### 2.2. Criteria and Definitions

AP was diagnosed when the patients presented with at least two of the following criteria [23,24]: (1) acute pancreatic abdominal pain; (2) serum lipase and/or amylase levels increased to three times the upper normal limit; (3) suitable imaging results (abdominal ultrasound, contrast-enhanced abdominal CT scanner, and/or magnetic contrast abdominal (MRI)). HTG-AP was diagnosed if AP patients presented with a blood triglyceride level >11.3 mmol/L and other causes were excluded [3,25].

### 2.3. Data Collection

We collected demographic data (age, gender, and comorbidities), clinical features, laboratory findings at admission (glucose, calcium, amylase, lipase, triglyceride, creatinine, aspartate aminotransferase, and alanine aminotransferase), the time between the onset of symptoms and hospitalization, and the severity of HTG-AP via standardized questionnaires. The severity of HTG-AP was assessed via the Balthazar grade, the SOFA, Ranson, and APACHE II scores, and the revised Atlanta classification [12,23].

### 2.4. Data Analysis

The data were double-entered using Microsoft Access, cleaned, and exported to STATA software version 17.0 (Copyright 1985–2021 StataCorp 4905 Lakeway Drive College Station, Texas 77845, TX, USA) for analysis. Continuous variables were analyzed and expressed as medians and ranges. Categorical variables were presented as numbers and proportions. Our main outcome was a decrease in serum triglycerides (<11.3 mmol/L) within 48 h of hospitalization. The main predictive factors associated with the outcome were demographic characteristics, comorbidities, the severity of disease, time between onset of symptoms and hospitalization, and treatment with TPE. A bivariable analysis was used to calculate the odds ratios (ORs) for the association between the decrease in serum triglycerides and the independent variables. Next, a multivariate analysis was adjusted for all variables with *p*-values < 0.2 in the univariate analysis. The multivariate analysis was performed using exact logistic regression. A predictive factor was considered statistically significant when the *p*-value was <0.05.

### 2.5. Ethical Approval

The protocol was approved by the institutional review board of Thai Binh University of Medicine and Pharmacy. The study was performed according to the good clinical practices recommended by the Declaration of Helsinki and its amendments. All patients or legal guardians of participants provided their written informed consent.

## 3. Results

### 3.1. Characteristics of Included Participants

During the study period, 81 patients with HTG-AP were hospitalized in the ICU and Gastroenterology Department. Of these patients, 30 received TPE, and the remaining 51 patients received conventional treatments.

The mean age of the participants was 45.3 ± 8.7 years. The majority of patients were male (82.7%) with a history of alcoholism (55.6%). Of the patients, 16.1% had diabetes.

There were no differences in age, gender, or comorbidities between the two groups of participants, the TPE and non-TPE groups (Table 1).

### 3.2. Participants’ Clinical Features and Laboratory Findings 

Abdominal pain was the most frequent clinical sign, occurring in all of 81 patients, followed by dyspepsia (87.7%), nausea or vomiting (72.8%), and a bloated stomach (61.7%). Overall, there were no differences in clinical symptoms between the two groups of patients except that those treated with TPE had higher rates of experiencing a bloated stomach and dyspepsia compared to non-TPE patients (Table 2).

Regarding the laboratory findings, the HTG-AP patients treated with TPE had significantly lower levels of calcemia and creatinemia but higher triglyceride levels than those in group that received conservative treatment (Table 2).

In both groups, most patients (63.0%) were admitted to the hospital early, within the first 24 h of the onset of the disease.

### 3.3. Severity of HTG-AP and Patient Outcomes 

The TPE patients had statistically significantly worse disease severity than the patients treated conservatively, as demonstrated by the Balthazar CT grade score ≥ D (93.3% vs. 72.6%, *p* = 0.02, respectively), Ranson score ≥ 3 (50.0% vs. 13.7%, *p* < 0.0001, respectively) and APACHE II score ≥ 8 (33.3% vs. 11.8%, *p* = 0.02, respectively). According to the revised Atlanta classification [23], severe AP was recorded in 43.4% of TPE patients, while the proportion was 15.7% in the non-TPE patients (Table 3).

All TPE patients were hospitalized in the ICU, while the rate of ICU hospitalization in the conservative treatment group was 5.9%. Additionally, the TPE patients had longer hospital stays than the non-TPE group.

However, the proportion of patients with a decrease in triglycerides <11.3 mmol/L after 48 h of treatment was higher in the TPE group than in the conservative treatment group (73.3% vs. 49.0%, *p* = 0.03, respectively).

Figure 1 shows the decrease in the blood triglycerides of the two groups of patients.

### 3.4. Predictive Factors Associated with Primary Outcome (Triglycerides < 11.3 mmol/L within 48 h) in HTG-AP Patients

The univariable and multivariable analyses showed that the decrease in triglyceride levels did not depend on the age, gender, or comorbidities of the HTG-AP patients or the severity of the disease. However, TPE and early treatment within the first 12 h of disease onset were effective at rapidly reducing the serum triglyceride levels (adjusted OR = 3.00, 95% CI = [1.04–8.43], *p* = 0.04 and aOR = 7.98, 95% CI = [1.44–44.14], *p* = 0.02, respectively) (Table 4).

## 4. Discussion

Our study showed the effectiveness of the TPE method in reducing triglyceride levels among patients with HTG-AP. The TPE patients achieved a reduction in triglycerides to <11.3 mmol/L after 48 h, a reduction which was three times higher than the group who did not receive the plasma exchange treatment. Moreover, the results of our study showed that the effectiveness of the TPE treatment was not influenced by the age, gender, comorbidities, or the severity of the AP. However, early treatment with TPE within the first 12 h of disease onset was found to be effective in rapidly reducing serum triglyceride levels.

Previous studies have shown that TPE treatment significantly reduces the burden of HTG-AP, including the patient mortality, length of stay in the ICU, and hospitalization rate. Although our study verified that early TPE provides better outcomes, prompt treatment with an appropriate strategy remains a challenge in the management of HTG-AP, especially in low- and middle-income countries. Reducing blood triglyceride levels plays an important role in the treatment of HTG-TP. Indeed, a strong positive interrelation between elevated triglyceride levels and the risk of severe AP was established well in previous studies [12,14,26,27,28,29,30]. Therefore, reducing triglyceride levels is crucial in preventing mortality due to HTG-AP [29]. HTG-AP patients exhibiting high triglyceride levels have higher rates of complications, organ failures, prolonged lengths of stay, and higher mortalities than those with pancreatitis without hypertriglyceridemia. A triglyceride level > 11.3 mmol/L is a known risk factor for AP.

Conventional treatment (fasting, lipid-lowering treatment, insulin, or fluid resuscitation) could gradually reduce triglyceride levels over a period of days to weeks [31]. In contrast, TPE provides a rapid removal of excess lipids from serum in about 2 h [32,33]. Previous studies reported the beneficial effect of TPE on HTG-AP [29,34]. However, there are currently limited reports on the effectiveness of this method compared to conventional treatment in HTG-AP patients. In our study, 28 out of 30 patients underwent TPE once, and only two patients received TPE twice during their hospital stay. The number of TPE sessions for a patient in this study was less than the number reported in other studies [14,35]. For instance, in a Vietnamese study, 93 TPE sessions were performed on 83 patients [14], while in a Hungarian study, 50 patients with HTG-AP received a total of 79 TPE sessions [35]. Nonetheless, our results also showed that blood triglyceride levels reached the therapeutic target after 48 h of TPE.

We found that the rate of triglyceride reduction was higher in patients receiving early TPE compared to those who received late treatment (>48 h after onset of disease). These findings emphasize the important role of early TPE in managing HTG-AP and reducing disease complications. The results of our study are consistent with previously published studies [12,14]. In a study conducted by Do et al. on 83 HTG-AP patients, early TPE sessions (<24 h) led to a rapid decrease in triglyceride levels compared to delayed treatment (>24 h after the onset of symptoms) [14]. Similarly, in a retrospective case–control study, Chen et al. evaluated the effects of TPE in 46 patients compared to conservative treatment in 135 other patients. TPE reduced serum triglycerides within 48 h, with the most significant results observed from the initiation of treatment initiation within 24 h; however, no benefit was observed after 72 h [12]. While TPE is more effective than conservative treatment in HTG-AP management, the triglyceride threshold for treatment indication is still unknown. Indeed, the triglyceride concentration that induces AP remains unclear as mild elevations of triglyceride levels can cause AP, and some patients with very severe hypertriglyceridemia do not develop the disease [36]. The incidence of pancreatitis varies with triglyceride levels, making it difficult to establish a single criterion for determining the initiation time of TPE [36].

In our study, most TPE patients were admitted to the ICU. This did not mean that TPE aggravated the severity of the disease, but rather that critically ill patients in the ICU were more likely to accept TPE than those in other wards. In fact, these patients were admitted to the ICU before receiving TPE. Our study also showed that the TPE patients had significantly worse diseases, and the lengths of their hospital stays were longer than those of the control group. However, only one TPE patient died, while two patients died in the non-TPE treatment group. Given the limited number of participants, this finding should be interpreted with caution. 

Our work has some limitations. Firstly, this was a single-center descriptive study with a modest sample size. Therefore, the results do not have enough statistical power for the findings to be extrapolated. In addition, the majority of the patients included were men with a mean age of 45 years; therefore, the results might not be representative of the general population. However, this feature is similar to the study conducted by Do et al. in Vietnam, which also had a majority of male participants with an average age of 42 years [13]. This could be explained by the sociodemographic characteristics of HTG-AP patients in Vietnam. In addition, CRP is of great importance as a single prognostic factor for AP. However, since CRP was not evaluated in many patients, we were unable to include it in our analysis. Another limitation of our study is that TPE indication was based entirely on patient consent and did not follow any randomization technique. Therefore, the possibility of selection or observation bias cannot be ruled out. Finally, we did not follow up with patients after their discharge from the hospital; therefore, the actual HTG-AP-related mortality has not been accurately assessed.

## 5. Conclusions

In conclusion, our results demontrate that the decrease in triglyceride levels among HTG-AP patients was independent of age, gender, comorbidities, or the severity of AP. This report highlights the effectiveness of early TPE in reducing triglyceride levels among HTG-AP patients. More randomized clinical trials with a large sample size and post-discharge follow-up are needed to confirm the effect of TPE on patient outcomes. Although early TPE offers better results in the treatment of HTG-AP, this treatment is expensive, requires modern equipment, and presents a challenge in clinical practice, especially in low- and middle-income countries. Therefore, future studies should evaluate the cost-effectiveness of TPE in the management of HTG-AP.

## Figures and Tables

**Figure 1 medicina-59-00864-f001:**
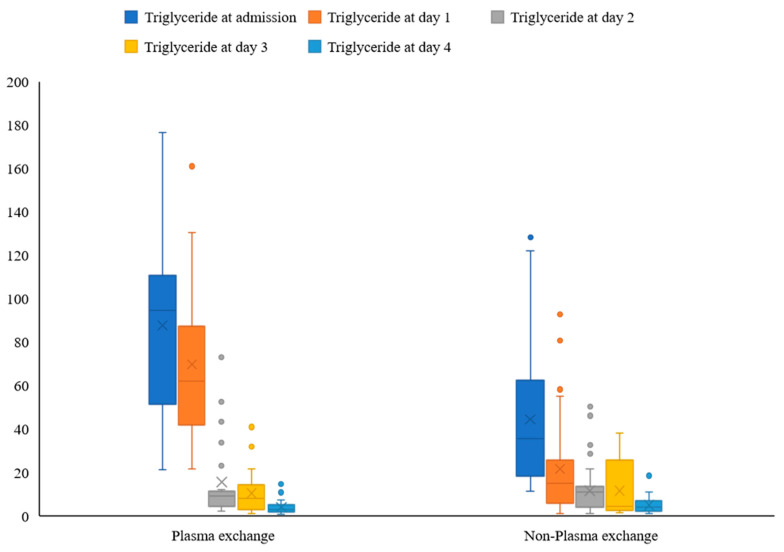
Effectiveness of TPE on decrease in triglycerides among HTG-AP patients.

**Table 1 medicina-59-00864-t001:** Characteristics of included patients.

	Non-TPE Group		TPE Group		Total		*p*-Value
	*n*	%	*n*	%	*n*	%	
Age (mean ± SD) (years)	45.4 ± 8.8		45.2 ± 8.7		45.3 ± 8.7		0.9
Age group							
<45	21	41.2	14	46.7	35	43.2	0.63
≥45	30	58.8	16	53.3	46	56.8	
Gender							
Female	9	17.7	5	16.7	14	17.3	0.91
Male	42	82.3	25	83.3	67	82.7	
Comorbidities							
Diabetes	10	19.6	3	10.0	13	16.1	0.26
History of pancreatitis	25	49.0	16	53.3	41	50.6	0.71
Alcoholism	27	52.9	18	60.0	45	55.6	0.54

TPE: therapeutic plasma exchange.

**Table 2 medicina-59-00864-t002:** Participants’ cinical features and laboratory findings.

	Non-TPE Exchange Group		TPE Group		Total		*p*-Value
	*n*	%	*n*	%	*n*	%	
Clinical features							
Fever	2	3.9	1	3.3	3	3.7	1.0
Rapid pulse	28	54.9	21	70.0	49	60.5	0.18
Abdominal pain	51	100	30	100	81	100	NA
Nausea, vomiting	38	74.5	21	70.0	59	72.8	0.66
Bloated stomach	25	49.0	25	83.3	50	61.7	0.002
Dyspepsia	42	82.4	29	96.7	71	87.7	0.06
Tenderness when touching the abdomen	2	3.9	5	16.7	7	8.6	0.10
Laboratory findings (bilan at admission)							
Glucose (mg/dL)	152.6 ± 82.1		184.9 ± 118.7		164.6 ± 97.8		0.15
Calci (mmol/L)	2.1 ± 0.2		1.8 ± 0.3		2.0 ± 0.3		<0.0001
Amylase (U/L)	263.3 ± 258.0		235.4 ± 296.2		245.5 ± 281.6		0.67
Lipase (U/L)	915.0 ± 958.7		988.9 ± 1001.3		942.4 ± 969.1		0.74
Triglyceride (mmol/L)	41.3 ± 30.2		87.5 ± 37.4		60.3 ± 39.0		<0.0001
Creatinine (mmol/L)	82.7 ± 28.2		70.3 ± 27.2		78.1 ± 28.3		0.03
AST (U/L)	124 ± 249.9		107.7 ± 80.4		118.0 ± 203.5		0.72
ALT (U/L)	63.8 ± 77.9		58.7 ± 88.8		61.9 ± 81.6		0.79
Time between onset and hospitalization (hours)							
≥48	5	9.8	5	16.7	10	12.3	0.77
24–<48	13	25.5	7	23.3	20	24.7	
12–<24	14	27.4	9	30.0	23	28.4	
<12	19	37.3	9	30.0	28	34.6	

ALT: alanine aminotransferase; AST: aspartate aminotransferase; TPE: therapeutic plasma exchange.

**Table 3 medicina-59-00864-t003:** Severity of hypertriglyceridemic pancreatitis and patient outcomes.

	Non-TPE Group		TPE Group		Total		*p*-Value
	*n*	%	*n*	%	*n*	%	
Severity of hypertriglyceridemic pancreatitis							
Balthazar CT grade							
<D	14	27.4	2	6.7	16	19.7	0.02
≥D	37	72.6	28	93.3	65	80.3	
SOFA							
<2	40	78.4	23	76.7	63	77.8	0.85
≥2	11	21.6	7	23.3	18	22.2	
Ranson score							
<3	44	86.3	25	50.0	59	72.8	<0.0001
≥3	7	13.7	15	50.0	22	27.2	
APACHE II							
<8	45	88.2	20	66.7	65	80.3	0.02
≥8	6	11.8	10	33.3	16	19.7	
Revised Atlanta classification [23]							
Mild AP	30	58.8	10	33.3	40	49.4	0.02
Moderately severe AP	13	25.5	7	23.3	20	24.7	
Severe AP	8	15.7	13	43.4	21	25.9	
Outcomes							
Length of stay	8.0 ± 2.8		10.7 ± 3.4		9.0 ± 3.3		0.0001
Transfer to intensive care unit	3	5.9	30	100.0	33	40.7	<0.0001
Death	2	3.9	1	3.3	3	3.7	1.0
Triglycerides < 11.3 mmol/L within 48 h	25	49.0	22	73.3	47	58.0	0.03

AP: acute pancreatitis; TPE: therapeutic plasma exchange.

**Table 4 medicina-59-00864-t004:** Predictive factors associated with primary outcome (triglycerides < 11.3 mmol/L within 48 h) in HTG-AP patients.

	Univariate Analysis			Multivariate Analysis		
	OR	95% CI	*p*-Value	Adjusted OR	95% CI	*p*-Value
Gender						
Female	reference					
Male	0.98	0.31–3.15	0.98			
Age group						
<45	reference					
≥45	0.97	0.40–2.37	0.96			
Comorbidities						
Diabetes	0.41	0.12–1.39	0.15	0.50	0.13–1.92	0.32
History of pancreatitis	0.77	0.32–1.86	0.57			
Alcoholism	0.89	0.37–2.17	0.80			
Severity of hypertriglyceridemic pancreatitis						
Balthazar CT grade						
<D	reference					
≥D	1.41	0.47–4.22	0.54			
SOFA						
<2	reference					
≥2	0.94	0.33–2.69	0.91			
Ranson score						
<3	reference					
≥3	1.48	0.54–4.05	0.45			
APACHE II						
<8	reference					
≥8	0.97	0.32–2.93	0.96			
Revised Atlantra classification [23]						
Mild AP						
Moderately severe AP	1.11	0.37–3.30	0.85			
Severe AP	0.81	0.28–2.35	0.70			
Time between onset and hospitalization (hours)						
≥48	reference			reference		
24–<48	2.85	0.57–14.32	0.20	3.44	0.61–19.29	0.16
12–<24	2.55	0.52–12.37	0.25	3.46	0.64–18.87	0.15
<12	5.83	1.20–28.37	0.03	7.98	1.44–44.14	0.02
Treatment						
Non-plasma exchange	reference			reference		
Plasma exchange	2.43	0.93–6.30	0.07	3.00	1.04–8.43	0.04

AP: acute pancreatitis.

## Data Availability

The data presented in this study are available upon reasonable request from the corresponding author (V.T.H.).

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
