# Peer review of "Effect of Plasma Exchange Treatment in Patients with Hypertriglyceridemia-Induced Acute Pancreatitis"

_medicina, 2023, doi:10.3390/medicina59050864_

Round 1

Reviewer 1 Report

Generally, the subject of this paper is noteworthy and has practical aspects.  However, similar single-center studies have been published. In my opinion, it is worth designing a good, multicenter study. We know that triglyceride levels decline as AP (acute pancreatitis)  is treated, so is it worth pursuing a procedure that offers only a slight advantage over conventional treatment?

Apart from that the following points required explanation and correction:

-In the introduction, the authors list hypertriglyceridemia in the first place, while it should be listed after gallstones and alcohol abuse

-In materials and methods, the authors do not mention CRP in the diagnosis of acute pancreatitis, but it is of great importance as a single prognostic factor

-The results said that 30 patients received TPE (therapeutic plasma exchange) and 51 received conventional treatment. It should be precisely described what conventional treatment was in question. Were the patients in the TPE group not receiving such treatment?

-Were mortality rates, and length of hospital stay compared between groups, or were the groups too small to draw such conclusions?

-The authors are right about the limitations of the study

There are a lot of language mistakes, all require correction.

I believe that the research paper submitted for referring, after completing all aforementioned corrections, may be published in Medicina. as another voice in the discussion on the use of TPE in AP.

Author Response

Generally, the subject of this paper is noteworthy and has practical aspects.  However, similar single-center studies have been published. In my opinion, it is worth designing a good, multicenter study. We know that triglyceride levels decline as AP (acute pancreatitis) is treated, so is it worth pursuing a procedure that offers only a slight advantage over conventional treatment?

Answer: Thank you very much for your comments. We agree with you that similar single-center studies have been published and multicenter studies are needed. However, as mentioned in the Introduction, in Vietnam, recent studies have shown that the incidence of acute pancreatitis (AP) is increasing with socio-economic development, especially hypertriglyceridemia-induced acute pancreatitis (HTG-AP). The method of plasma exchange (TPE) in the HTG-AP treatment is being applied in many central and regional hospitals but the effectiveness of this treatment is still unknown in the country. With limited economic conditions, our single-center observational study showed preliminary results. In addition to confirming the effectiveness of TPE in the treatment of HTG-AP, our results also indicate that the effectiveness of TPE treatment is independent of age, sex, comorbidities or severity of AP, but rather depends on the initiation of treatment since onset of symptoms.

We also agree with you that triglyceride levels decrease when AP is treated. However, we believe that it is worthwhile to evaluate the therapeutic effect of TPE method compared with conventional treatment in AP. Indeed, as mentioned in the Discussion section, conventional treatment (fasting, lipid-lowering treatment, insulin, or fluid resuscitation) could gradually reduce triglyceride over a period of days to weeks. In contrast, TPE provides rapid removal of excess lipids from serum in about 2 hours. Reducing blood triglyceride levels plays an important role in the treatment of HTG-TP. HTG-AP patients exhibiting high triglyceride levels have higher rates of complications, organ failure, prolonged length of stay, and higher mortality than those with pancreatitis without hypertriglyceridemia.

Apart from that the following points required explanation and correction:

-In the introduction, the authors list hypertriglyceridemia in the first place, while it should be listed after gallstones and alcohol abuse

Answer: Thank you very much for your suggestions. We corrected as:

“Gallstones and alcoholism are known to be the leading causes of AP. However, hypertriglyceridemia is also an important and increasingly common cause”

-In materials and methods, the authors do not mention CRP in the diagnosis of acute pancreatitis, but it is of great importance as a single prognostic factor

Answer: Since CRP was not evaluated in many patients, we were unable to include it in the analysis. However, we also added this point to the Limitations of the study as:

“In addition, CRP is of great importance as a single prognostic factor for AP. However, Since CRP was not evaluated in many patients, we were unable to include it in our analysis.”

-The results said that 30 patients received TPE (therapeutic plasma exchange) and 51 received conventional treatment. It should be precisely described what conventional treatment was in question. Were the patients in the TPE group not receiving such treatment?

Answer: Apologies for our mistake. Apart from plasma exchange, TPE patients received also conventional treatment. We corrected it in the Method section as:

“If they agreed, they were classified in the TPE group (in addition to plasma exchange, these patients also received conventional treatment), otherwise, they were placed in the non-TPE group (conventional treatment only)”

-Were mortality rates, and length of hospital stay compared between groups, or were the groups too small to draw such conclusions?

Answer: Mortality rate and length of hospital stay have been analyzed and presented (please refer the Table 3). This point has been also discussed as:

“In our study, most TPE patients were admitted to the ICU. This did not mean that TPE aggravated the severity of disease, but rather that critically ill patients in the ICU were more likely to accept TPE than those in other wards. In fact, these patients were admitted to the ICU before receiving TPE. Our study also showed that the TPE patients had significantly worse diseases and their length of hospital stay was longer than that of the control group. However, only one TPE patient died, while the other two died in the non-TPE treatment group. This finding should also be interpreted with caution, given the limited number of patients.”

-The authors are right about the limitations of the study

Answer: Thank you very much.

There are a lot of language mistakes, all require correction.

Answer: The language was now checked by professional and native English-speaking.

I believe that the research paper submitted for referring, after completing all aforementioned corrections, may be published in Medicina. as another voice in the discussion on the use of TPE in AP.

Answer: Once again, thank you very much for your valuable comments and suggestions.

Reviewer 2 Report

This singe center study aimed to evaluate the effectiveness of therapeutic plasma exchange in hypertrigliceridemia induced acute pancreatitis in Vietnam.

-It is a well written article, hovewer the study design was not a randomized study and all of the TPE patients were admitted to ICU, which could be an important outcome parameter as well.

-The cost-effectiveness would be also important, but in this case, as all of the TPE patients were cared at ICU, it would cause a bias as well.

-The main problem is, that that the severity of the disease was based on the predicting scores. It should be made based on the modified Atlanta classification. This part should be reworked.

-This study cannot answer the question, which therapeutic modality is better, but a randomized study, as the ELEFANT trial will maybe show it and guide the patient management.

Author Response

This singe center study aimed to evaluate the effectiveness of therapeutic plasma exchange in hypertrigliceridemia induced acute pancreatitis in Vietnam.

Answer: Thank you very much for your valuable comments and suggestions.

-It is a well written article, however the study design was not a randomized study and all of the TPE patients were admitted to ICU, which could be an important outcome parameter as well.

Answer: Thank you very much for your remark. This point has been discussed in the Limitation section.

-The cost-effectiveness would be also important, but in this case, as all of the TPE patients were cared at ICU, it would cause a bias as well.

Answer: The fact that all of the TPE patients were treated at ICU was discussed as a limitation of our study. We also added the cost-effectiveness point in the Conclusion as:

“Although early TPE offers better results in the treatment of HTG-AP, this treatment is expensive, requires modern equipment, and presents a challenge in clinical practice, especially in low- and middle-income countries. Therefore, future studies should evaluate the cost-effectiveness of TPE in HTG-AP management.”

-The main problem is, that that the severity of the disease was based on the predicting scores. It should be made based on the modified Atlanta classification. This part should be reworked.

Answer: Thank you very much for your comment and suggestion. We revised our analysis by adding the severity of AP according to the revised Atlanta classification. The conclusion of our study was finally unchanged. Especially:

TPE patients had statistically significantly worse disease severity than patients treated conservatively. According to the revised Atlanta classification, severe AP was recorded in 43.4% TPE patients, while the proportion was 15.7% in non-TPE patients, moderately severe AP was reported in 23.3% and 25.5% and mild AP was in 33.3% and 58.8%, respectively (Please refer to Table 3). However, the univariable and multivariable analysis showed that the decrease in triglyceride did not depend on age, gender, comorbidities of HTG-AP patients, or severity of disease (Please refer to Table 4).

Reference:

Banks PA, Bollen TL, Dervenis C, Gooszen HG, Johnson CD, Sarr MG, Tsiotos GG, Vege SS; Acute Pancreatitis Classification Working Group. Classification of acute pancreatitis--2012: revision of the Atlanta classification and definitions by international consensus. Gut. 2013 Jan;62(1):102-11. doi: 10.1136/gutjnl-2012-302779.

-This study cannot answer the question, which therapeutic modality is better, but a randomized study, as the ELEFANT trial will maybe show it and guide the patient management.

Answer: We agree with you that our study cannot confirm which treatment is better because it is simply a single-center descriptive study with a modest sample size.  This has been discussed in the Limitations section. However, our results showed that TPE method is effective in reducing triglyceride among patients admitted with HTG-AP, three times higher compared to the group who did not receive plasma exchange. Moreover, our findings suggest that the effectiveness of TPE treatment is independent of age, sex, comorbidities or severity of AP, but rather depends on the initiation of treatment since onset of symptoms. We also discussed in the Conclusion section that more randomized clinical trial studies with a large sample size and post-discharge follow-up are needed to confirm the effect of TPE on patients’ outcomes.

Round 2

Reviewer 2 Report

I accept the canges.